# Efficient Object Manipulation Planning with Monte Carlo Tree Search

Huaijiang Zhu[1], Ludovic Righetti[1,2]

*Abstract*— This work presents an efficient approach to object manipulation planning using Monte Carlo Tree Search (MCTS) to find contact sequences and an efficient ADMM-based trajectory optimization algorithm to evaluate the dynamic feasibility of candidate contact sequences. To accelerate MCTS, we propose a methodology to learn a goal-conditioned policy-value network used to direct the search towards promising nodes. Further, manipulation-specific heuristics enable to drastically reduce the search space. Systematic object manipulation experiments in a physics simulator demonstrate the efficiency of our approach. In particular, our approach scales favorably for long manipulation sequences thanks to the learned policy-value network, significantly improving planning success rate.

## I. INTRODUCTION

The ability to plan sequences of contacts and movements to manipulate objects is central to endow robots with sufficient autonomy to perform complex tasks. This remains, however, particularly challenging. Indeed, finding dynamically feasible sequences of contacts between the manipulator and an object typically leads to intractable combinatorial and nonlinear problems.

Recently, trajectory optimization has become a popular tool for multi-contact locomotion [1]–[4] as this leads to desirable formulations to reason about interaction forces. Yet, it remains unclear how the planning of contact modes should be efficiently incorporated, primarily due to its discrete nature which creates an undesirable consequence: discontinuity in the dynamics at contact switch. To handle this discontinuity under the trajectory optimization framework, two streams of methodologies have emerged:

1) the contact-invariant or contact-implicit approach enforce contact complementarity either as hard constraints [5], [6], penalty terms in a cost function [7]–[9], or with differentiable soft contacts models [10]
2) the hybrid approach treats contact switches as discrete decisions within a continuous problem [11]–[13].

In this work, we examine the latter methodology to propose an optimization framework amenable to customization with object manipulation-specific heuristics and learning from data to improve its computational efficiency.

The most common formulation of such problem is via Mixed-integer Programming (MIP). In the context of robot manipulation, one representative work is the Contact-Trajectory Optimization proposed in [12], where contact scheduling is modeled as binary decision variables and the non-convexity due to cross product is relaxed by McCormick envelopes. The resulted problem is a Mixed-integer

Quadratic Program (MIQP) which can be solved by off-the-shelf MIQP solvers. However, the approach has only been demonstrated on 2D object manipulation with very short manipulation sequences. This is in contrast to our approach which handles 3D objects and long sequences.

In the context of machine learning, CoCo proposed in [14] finds feasible solution to MIP by first learning to map the problem parameters to the assignment of the discrete variables offline and then solving the resulted continuous optimization problem online. While this greatly improves the solution speed at inference time, it assumes that one is able to solve the original MIP in a reasonable amount of time to construct the dataset. If the original problem is prohibitive to solve, collecting a large dataset for this problem may not be practical without abundant computational resources.

Recently, an algorithm that augments Contact-Implicit Trajectory Optimization (CITO) with tree search was proposed in [15] to incorporate domain-specific knowledge for robot manipulation. It uses depth-first search (DFS) to find a sequence of kinematically feasible contact modes that have a stable grasp and then constrain the CITO problem with the found contact sequence.

In principle, we can employ a brute-force approach to our problem: search over all possible combinations of the discrete variables and for each such combination solve the resulted continuous optimization problem. In general, such strategy is not practical due to the factorial complexity. However, it can be made more efficient if 1) the search space can be notably reduced, 2) good search heuristics are available, and 3) the non-convex continuous optimization problem can be solved efficiently. In this work, we show that all these three requirements can be achieved. In particular, our contributions are

1) we adapt learning-based Monte Carlo Tree Search (MCTS) to discrete contact planning problems for robotic manipulation,
2) we formulate the resulted continuous optimization problem as a biconvex program to allow efficient solution via the Alternating Direction Method of Multipliers (ADMM) [16], and
3) we learn a policy-value network from data collected on short-horizon tasks which provides good heuristics for long-horizon tasks and significantly decreases the overall solution time.

[1]Tandon School of Engineering, New York University, USA
[2]Max-Planck Institute for Intelligent Systems, Germany

An extended version of this work has been submitted to IROS 2022 and is under review.

To our best knowledge, this is the first application of learning-based MCTS to contact planning for manipulation.

## II. PROBLEM STATEMENT

### A. Inputs

We aim to solve an object manipulation task similar to the Contact-Trajectory Optimization problem proposed in [12] where the following quantities are given:

1) a rigid object with known geometry, friction coefficient $\mu$, mass $m$, moment of inertia $\mathcal{I}$, and $N_\Omega$ pre-defined touchable regions,
2) a trajectory with discretization step $\Delta t$ of length $T$ that consists of the desired object pose, velocity, and acceleration
3) an environment with known geometry and friction coefficient $\mu_e$, and
4) a manipulator with known kinematics that can make at most $N_c$ contacts with the object.

At the $t$-th time step, given the object motion and the object dynamics, we can compute the desired total force $f_{\text{des}}(t)$ and torque $\tau_{\text{des}}(t)$ to be applied to the object from rigid-body dynamics. In addition, as the geometry of the object and the environment as well as the object motion are known, we can obtain $N_e(t)$ environment contact locations $r_e(t)$ for $e \in \{1, \ldots, N_e(t)\}$ at each time step $t$ by checking the collisions between the object and the environment, assuming uniform pressure distribution.

### B. Outputs

For each time step $t$, we aim to find the following:

1) the contact region $\Omega_c(t) \in \{0, 1, \ldots, N_\Omega\}$, the contact force $f_c(t)$ and the contact location $r_c(t)$ for each contact point $c$ of the manipulator; $\Omega_c(t) = 0$ indicates that the $c$-th contact point is not in contact, and
2) the environment contact force $f_e(t)$

such that the forces and torques sum to the desired ones

$$\sum_{c=1}^{N_c} f_c(t) + \sum_{e=1}^{N_e(t)} f_e(t) = f_{\text{des}}(t) \qquad (1a)$$

$$\sum_{c=1}^{N_c} r_c(t) \times f_c(t) + \sum_{e=1}^{N_e(t)} r_e(t) \times f_e(t) = \tau_{\text{des}}(t). \qquad (1b)$$

## III. METHOD

The problem described above is challenging even though the desired object motion is provided, as one needs to find both the discrete contact region $\Omega_c(t)$ and the continuous manipulator contact forces $f_c(t)$, the contact locations $r_c(t)$, and the environment contact forces $f_e(t)$.

### A. Discrete Contact Planning via MCTS

A series of learning-based MCTS algorithms has been proposed in [17], [18] for the chess-playing agents AlphaGo and AlphaZero. We adapt it to solve the discrete contact region planning problem and refer the algorithm as Policy-Value Monte Carlo Tree Search (PVMCTS): for a given object motion $\xi$, we want to find the contact region for each contact point $c$ at each time step $t$. The whole sequence is then evaluated to return a reward $r$ to guide future search.

*1) Assumptions:* To reduce the search space, we make the following assumptions:

- **Persistent contact:** While the downstream continuous optimization problem may have a small discretization step, for example $\Delta t = 0.1\,\text{s}$, most manipulation tasks do not require contact switch at such a high frequency. Thus, we assume that a contact point must remain in the same region for $d$ time steps.
- **Contact switch:** We allow at most one contact point to break or make contact at each contact switch, and we only allow contact switches when the desired object velocity and acceleration are zero.
- **Contact region:** Each contact region can only be touched by at most one contact point.

*2) State and action representation:* With the assumption above, at each planning step $n$, the PVMCTS chooses for each contact point $c$ its contact region for the next $d$ time steps, hence $a_n = [\Omega_1(t), \ldots, \Omega_{N_c}(t)]_{t=nd}^{nd+d}$, and the state $s_n$ is simply the concatenation of all previous actions.

*3) Heuristics:* To reduce the search space, we apply the following heuristics to further limit the size of the legal action set $\mathcal{A}(s)$:

- **Kinematic feasibility:** For each contact point $c$, a contact region will only be considered if inverse kinematics can find a manipulator configuration that reaches the center of this region within an error threshold of $1\,\text{cm}$.
- **Number of contacts:** For time steps where the angular acceleration is nonzero, we require at least $\min(N_c, 3)$ contact points to be in contact.

*4) Reward function:* Once the PVMCTS reaches a terminal state, hence $Nd = T$, we obtain a sequence of contact regions $[\Omega_1(t), \ldots, \Omega_{N_c}(t)]_{t=0}^{T-1}$, which is used to construct a continuous optimization problem that solves for the contact force $f_c(t)$, $f_e(t)$ and the contact location $r_c(t)$ (cf. Sec III-B). To evaluate this solution, we integrate it to obtain an object pose $\hat{q}(T-1)$ with the semi-implicit Euler method. We then compare it with the desired pose $q(T-1)$ to compute a weighted distance

$$D(q, \hat{q}) = \|p - \hat{p}\| + \beta \left\| \log(\hat{R}^{\mathsf{T}} R) \right\|, \qquad (2)$$

where $\beta > 0$ scales the angular distance. The weighted distance within a threshold $D \leq D_{\text{th}}$ is then normalized to $[0, 1]$ to obtain the reward.

*5) Goal-conditioned policy-value network:* Note that each PVMCTS instance only searches for the contact sequence for a given object motion $\xi$, thus the rewards are motion-specific. To allow learning from object motion information as well, we define an intermediate goal $\lambda_n = [q(nd), q(nd + h)]$ for each planning step $n$ that consists of the current desired object pose $q(nd)$ and the future one $q(nd + h)$ in $h$ steps. Fig 1 depicts the policy-value network architecture.

*6) Value classifier:* One key difference between our task and the generic game-play is that our dataset is highly imbalanced—many contact sequences explored by the PVMCTS are dynamically infeasible, resulting in rewards that equal zero. Directly training on such a dataset leads to underestimation of the value function. Instead, we only

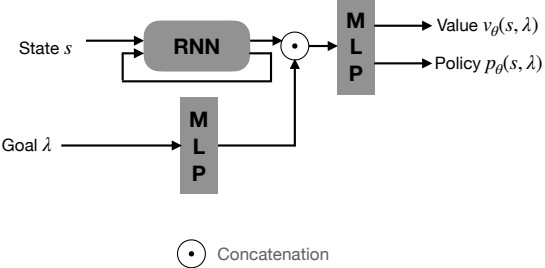

State s

RNN

Value $v_\theta(s, \lambda)$

Policy $p_\theta(s, \lambda)$

Goal $\lambda$

MLP

MLP

⊙ Concatenation

Fig. 1: Schematic diagram of the policy-value network architecture. Activation functions and regularization layers such as Dropout and BatchNorm are omitted.

train our policy-value network on samples that incur positive rewards. Additionally on the entire dataset $\mathcal{D}$, we train a binary classifier $C_\phi(s)$ with logistic regression where positive samples are given more weights. At inference time, a state is only fed into the policy-value network if the classifier labels it as positive; otherwise, we simply output zero value $v_\theta(s) = 0$ and uniformly distributed action probability $p_\theta(s, a) = \frac{1}{|\mathcal{A}(s)|}$.

### B. Continuous Contact Optimization via ADMM

Now let us consider the sub-problem where we already obtained a sequence of contact regions $[\Omega_1(t), \dots, \Omega_{N_c}(t)]_{t=0}^{T-1}$ for each contact point $c$: we can find the contact forces and locations by solving a continuous optimization problem with the following constraints and cost. For brevity, we omit the time indices if there is no ambiguity.

*1) Dynamics:* The contact forces and torques must sum to the desired ones

$$\sum_{c=1}^{N_c} f_c + \sum_{e=1}^{N_e} f_e = f_{\text{des}} \tag{3a}$$

$$\sum_{c=1}^{N_c} r_c \times f_c + \sum_{e=1}^{N_e} r_e \times f_e = \tau_{\text{des}}. \tag{3b}$$

*2) Contact location:* The contact location must be inside the given contact region $\Omega_c$ for $\Omega_c \neq 0$.

*3) Contact force:* If the $c$-th contact point is not in contact with any contact region, hence $\Omega_c = 0$, the contact force is set to zero. Note that this is not a complementarity constraint as $\Omega_c$ is already given.

*4) Sticking contact:* If the $c$-th contact point is in contact with the same region at two consecutive time steps, then the contact location remains the same to prevent the manipulator from sliding on the object.

*5) Coulomb friction:* The contact force has to stay inside the friction cone of the given surface. Note that the environment contact can be either sticking or sliding depending on the velocity of the contact point $\dot{r}_e(t)$ relative to the environment, which can be obtained from the object motion.

*6) Performance cost:* Finally, we minimize a quadratic objective function that avoids applying large forces at the boundary of the contact region

$$J = \sum_{t=0}^{T-1} \sum_{c=1}^{N_c} \|f_c(t)\|^2 + \|r_c(t)\|^2 \tag{4}$$

*7) Biconvex Decomposition:* The continuous optimization problem described above has an interesting feature that the only non-convex constraint (3b) due to the cross product $r_c \times f_c$ is in fact biconvex. When we group the decision variables into two sets $x = [r_c(t), \alpha_c(t)]_{t=0}^{T-1}$ and $z = [f_c(t), f_e(t)]_{t=0}^{T-1}$, we can re-write the original problem into the standard ADMM form with a biconvex constraint

$$G(x, z) = \sum_{c=1}^{N_c} r_c \times f_c + \sum_{e=1}^{N_e} r_e \times f_e - \tau_{\text{des}} = 0. \tag{5}$$

As all other constraints are separable in $x$ and $z$, they can be added as indicator functions to the objective and at each ADMM update step solved as standard constrained Quadratic Programs (QPs).

## IV. EXPERIMENTS

We conduct simulation experiments to show that our framework 1) is capable of finding dynamically feasible solutions to manipulation planning problems defined in Sec II, and 2) scales to long-horizon tasks even when trained only on data collected from shorter-horizon tasks.

### A. Experiment Setup

Throughout all experiments, we consider a manipulator with $N_c = 2$ contact points, composed of two modular robot fingers similar to the ones used in [19] and a $10\,\text{cm} \times 10\,\text{cm} \times 10\,\text{cm}$ cube with mass $m = 0.5\,\text{kg}$ on an infinitely large plane. The cube and the plane have the same friction coefficient $\mu = \mu_e = 0.8$. We consider the following primitive object motions and the composite of them 1) Sliding (*S*) 2) Sliding with curvature (*SC*) 3) Rotating (*R*) 4) Lifting (*L*), and 5) Pivoting (*P*) generated by interpolating between the initial and desired object poses. An interpolated trajectory for a single primitive motion has $T = 48$ time steps and lasts $4.8\,\text{s}$. We require each contact point remain in the same region for $d = 8$ time steps, hence the trajectory has $N = 6$ contact modes. The trajectory always starts with zero velocity and acceleration for $2.4\,\text{s}$ allowing at most two contact switches.

### B. Metrics

We examine three performance metrics to evaluate the effectiveness and efficiency of our method

1) **Pose error:** the error between the desired pose and the one integrated from the solution.
2) **Number of evaluations:** the number of continuous optimization problems the PVMCTS needs to solve until it finds the first feasible solution below the error threshold $D_{\text{th}}$.
3) **Solution time:** the total time needed to find the first feasible solution.

TABLE I: Task performance for motions interpolated from randomly sampled poses with various lengths. Pose errors are calculated only for successful tasks.

| # Object motions | Trajectory length $T$ | Model | Success rate | Error [cm,°] Average | Worst | # Evaluation Average | Worst | Time [s] Average | Worst |
|---|---|---|---|---|---|---|---|---|---|
| 1 | 48 | Untrained | **20/20** | $0.16, 1.18$ | $0.57, 5.89$ | 4.65 | 11 | 2.09 | 4.88 |
| | | Trained | **20/20** | $\mathbf{0.15, 0.39}$ | $\mathbf{0.24, 0.83}$ | **1.5** | **4** | **0.71** | **1.73** |
| 2 | 96 | Untrained | **20/20** | $0.35, 1.23$ | $0.79, 2.24$ | 8.15 | 25 | 8.54 | 21.88 |
| | | Trained | **20/20** | $\mathbf{0.32, 0.88}$ | $\mathbf{0.48, 1.78}$ | **2** | **5** | **1.96** | **4.68** |
| 3 | 144 | Untrained | 12/20 | $0.48, 1.86$ | $0.91, 5.98$ | 29.85 | 50 | 46.23 | 84.63 |
| | | Trained | **20/20** | $\mathbf{0.43, 1.81}$ | $\mathbf{0.58, 4.84}$ | **2.3** | **8** | **3.18** | **9.43** |
| 4 | 192 | Untrained | 5/20 | $\mathbf{0.61, 1.95}$ | $\mathbf{0.74, 2.13}$ | 43.05 | 50 | 93.57 | 137.31 |
| | | Trained | **20/20** | $0.65, 2.56$ | $1.59, 6.92$ | **2.8** | **16** | **6.12** | **31.02** |

## C. Untrained PVMCTS

In this set of experiments, we show that our method is capable of generating feasible contact plans for primitive object motions using an untrained PVMCTS. The network outputs are simply set to $v_\theta(s, a) = 0$ and $p_\theta(s, a) = \frac{1}{|\mathcal{A}(s)|}$.

*1) Tasks:* In this experiment, we consider for each primitive object motion the following desired poses summarized in Table II. They are given relative to the initial object pose and the orientation is expressed in the axis-angle representation.

TABLE II: Desired object poses for various primitive motions.

| Tasks | Position [cm] | Orientation [°] |
|---|---|---|
| $S$ | $[0, 10, 0]$ | $[0, 0, 0]$ |
| $SC$ | $[0, 5, 0]$ | $[0, 0, 45]$ |
| $R$ | $[0, 0, 0]$ | $[0, 0, 90]$ |
| $L$ | $[0, 0, 10]$ | $[0, 0, 0]$ |
| $P$ | $[5, 0, 2]$ | $[0, 45, 0]$ |

*2) Results:* Table III shows that our method, even with an untrained model, is capable of finding dynamically feasible solutions for object motions after only a handful evaluations on average. Indeed, the heuristics we proposed greatly reduces the search space while still allowing discovery of dynamically feasible contact plans that results in small pose errors for the object motions considered in this task.

*3) Executing the contact plan:* To validate the solution found by ADMM, we execute the contact plan in an open-loop fashion with a simple impedance controller for each finger in the PyBullet simulator [20]. In simulation, the robot is able to move the object towards its desired pose even without the feedback of the actual object pose.

TABLE III: Task performance for primitive object motions.

| Tasks | Error [cm,°] Average | Worst | # Evaluation Average | Worst | Time [s] Average | Worst |
|---|---|---|---|---|---|---|
| $S$ | $0.25, 0.72$ | $0.27, 1.69$ | 6.3 | 15 | 3.10 | 7.03 |
| $SC$ | $0.09, 0.42$ | $0.14, 0.42$ | 4.3 | 12 | 2.31 | 6.38 |
| $R$ | $0.00, 1.69$ | $0.00, 1.69$ | 2.5 | 9 | 1.16 | 3.96 |
| $L$ | $0.35, 0.00$ | $0.35, 0.00$ | 5.7 | 12 | 2.48 | 4.89 |
| $P$ | $0.36, 2.12$ | $0.42, 3.84$ | 8.2 | 19 | 3.62 | 7.56 |

## D. Learning Planar Manipulation Tasks

In the previous experiments, we have shown the effectiveness of our search strategy thanks to the heuristics that greatly reduces the size of the set of legal actions $\mathcal{A}(s)$.

Nevertheless, the search space still grows exponentially with the length of the contact sequence $N$. It is thus natural to ask if learning from past experience can accelerate the search.

In this experiment, we show that we can significantly reduce the solution time for longer-horizon tasks even if they are not contained in the training data.

*1) Tasks:* : We consider the composition of planar object motions *SC* with randomly sampled desired poses, which are then interpolated. The trajectories in the training data all have a length of $T = 96$.

*2) Results:* We evaluate the trained and untrained models on tasks that are generated by the same procedure yet have different trajectory lengths. Each task category with the same trajectory length has 20 different randomly generated tasks. We set the maximal number of evaluations to be 50, hence a task is considered failed if no feasible solution within the error threshold is found after evaluating 50 contact sequences. Table I reports the performance metrics of the untrained and trained model for each task category. We see that the trained model consistently solve all the tasks, regardless of the trajectory length, while the untrained model struggles in long-horizon tasks, solving only 5 out of 20 tasks with trajectory length $T = 192$. In contrast to the untrained model, the average number of evaluations required by the trained model to find the first feasible solution grows rather slowly with the trajectory length.

## V. CONCLUSION

In this work, we proposed a framework that combines data-driven tree search via PVMCTS and efficient non-convex optimization via ADMM to find dynamically feasible contact forces and locations to realize a given object motion. We show that the capability of learning from data allows our framework to achieve great scalability for long-horizon motions even when the dataset only contains data collected from shorter motions.

The most limited aspect of our approach is that the object motion must be provided. While this is possible for simple tasks, true dexterity requires automatic generation of the object motion by reasoning about the environment, which can be achieved by enumerating not only the manipulator contacts but also the environment contacts as proposed in [21]. It is thus an interesting future research direction to incorporate such a component.

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
