# OpenReview forum: "Efficient Object Manipulation Planning with Monte Carlo Tree Search"
_ICRA.org/2022/Workshop/Contact-Rich — ICRA 2022 Workshop: RL for Manipulation Poster_

### Official Review · Reviewer_FSX3 · 2022-05-05
**Interesting paper that aims to make learning based MCTS more efficient, needs some adjustment to be able to stand as a stand-alone short paper**

**Rating:** 7
**Confidence:** 3

**Review:**

This paper proposes a few modifications to make MCTS more efficient for the challenging problem of contact planning. More specifically, the authors choose a set of reasonable assumptions about the contacts to minimize the search space, require a minimum amount of 3 contact points at all times, as well as filter out states through beneficial configurations based on Inverse Kinematics. Lastly to avoid biasing their value function, they make sure to train only on contact sequences that result in positive reward and learn a binary classifier to aid with that during inference.
The paper is interesting and the results suggest that with these modifications the algorithm scales well for longer planning horizons. I do however have some concerns that are probably related to the fact that the paper needed to be revised to a shorter format.

1. The challenging nature of this specific problem could have been motivated further in the introduction. Why a basic MCTS is insufficient for this kind of problems? What uncertainties/challenges is the learning based solution addressing given the known information about the object and its motion?

2. How exactly is the binary classifier used during inference? Why is it necessary?

3. I think that in Table 1 it would be beneficial to also report the percentual improvement between the trained and untrained versions for each motion and trajectory length. On the same note, it would be very interesting to comment on the better performance of the untrained model for the longest trajectory length.

4.  In Section IV.C.3, you are also referring to the execution of the contact plan but there are no results pertaining to it, which I am assuming is due to the lack of space. In that case, maybe consider removing this subsection.

5. For your future work, it would be very interesting to see what sensitivity your algorithm has to modelling uncertainties about the object, given the rather strict assumptions about its physical properties and motion.

In conclusion, this paper proposes a way of making learning-based MCTS applicable to contact planning that scales well with longer trajectory lengths. The paper is well-written but may need some adjustments to make it self-contained in this format.

---

### Official Review · Reviewer_qBi3 · 2022-05-12
**Efficient object manipulation planning - review**

**Rating:** 8
**Confidence:** 4

**Review:**

The authors develop a method to sequence discrete contacts plans and learning a policy-value network that can provide useful heuristics for long-horizon manipulation trajectories. Their network additionally accelerates MCTS by identifying nodes in the tree that are more worth exploring than others, allowing their method to scale to long-horizon tasks. They call this method PVMCTS, and empirically show it's performance on a modular two-fingered robot benchmark task which manipulates a cube using 5 different primitive motions. One unique take-away from their approach is how they balance the dataset of trajectories to train the policy network by only including trajectories with positive reward signals, as much of the exploratory policy's trajectories include no useful information in training the value function (resulting in underestimation). Additionally, they show that without training the value network, the model can only complete 5 of the 20 tasks, all of which are solved by their trained model. One aspect of their experimental setup that is a bit mysterious is how they represent the object motions referred to in Table I. Their main claim that their method improves the baseline performance on longer horizon tasks is well-supported, and might be worth scaling and comparing to pure-learning based methods on longer trajectories in a future iteration of this approach. Lastly, the training details of their value network are a bit sparse/missing from the main text.